# BMP6 Promotes the Secretion of 17 Beta-Estradiol and Progesterone in Goat Ovarian Granulosa Cells

**DOI:** 10.3390/ani12162132

**Published:** 2022-08-19

**Authors:** Shuaifei Song, Wenfei Ding, Hui Yao, Lei Wang, Bijun Li, Yukun Wang, Xue Tang, Yiyu Zhang, Deli Huang, Dejun Xu, Zhongquan Zhao

**Affiliations:** 1Chongqing Key Laboratory of Herbivore Science, College of Animal Science and Technology, Southwest University, Chongqing 400715, China; 2Tengda Animal Husbandry Co., Ltd., Chongqing 402360, China

**Keywords:** BMP6, goat ovarian granulosa cells, estradiol, progesterone

## Abstract

**Simple Summary:**

As a member of the bone morphogenetic protein families (BMPs), BMP6 is a key regulatory factor in the ovaries. It is expressed in the ovarian granulosa cells (GCs) of different species and plays an important regulatory role in follicular growth and development. Moreover, the steroid hormones secreted by ovarian granulosa cells are involved in important physiological processes such as follicle development, ovulation, and embryo implantation. However, the role of BMP6 in steroid hormone synthesis in goat ovarian GCs remains unclear. We aimed to examine the effects of BMP6 on the function of goat ovarian GCs. The results showed that BMP6 did not significantly affect the proliferation, cell cycle, and apoptosis of ovarian GCs but it up-regulated the expression of the steroid hormone synthesis rate-limiting enzymes *CYP19A1* and *CYP11A1* to promote the production of 17 beta-estradiol (E2) and progesterone (P4). This finding provides a new idea for the study of steroid hormone synthesis in follicles.

**Abstract:**

The purpose of this study was to investigate the effects of BMP6 on the function of goat ovarian granulosa cells (GCs). The results showed that the exogenous addition of BMP6 did not affect the EdU-positive ratio of ovarian GCs and had no significant effect on the mRNA and protein expression levels of the proliferation-related gene *PCNA* (*p* > 0.05). Meanwhile, BMP6 had no significant effect on the cycle phase distribution of GCs but increased the mRNA expression of *CDK4* (*p* < 0.05) and *CCND1* (*p* < 0.01) and decreased the mRNA expression of *CCNE1* (*p* < 0.01). Moreover, BMP6 had no significant effect on the apoptosis rate of GCs and did not affect the mRNA expression levels of apoptosis-related genes *BAX*, *BCL2,* and *Caspase3* (*p* > 0.05). Importantly, BMP6 upregulated the secretion of 17 beta-estradiol (E2) and progesterone (P4) in ovarian GCs (*p* < 0.01). Further studies found that BMP6 inhibited the mRNA expression of *3β-HSD* and steroid synthesis acute regulator *(StAR)* but significantly promoted the mRNA expression of the E2 synthesis rate-limiting enzyme *CYP19A1* and the P4 synthesis rate-limiting enzyme *CYP11A1* (*p* < 0.01). Taken together, these results showed that the exogenous addition of BMP6 did not affect the proliferation, cell cycle, and apoptosis of goat ovarian GCs but promoted the secretion of E2 and progesterone P4 in ovarian GCs by upregulating the mRNA expressions of *CYP19A1* and *CYP11A1*.

## 1. Introduction

Follicles are the basic unit of the ovary that exert biological functions, and the development of follicles is the main factor that determines reproductive performance in female animals. The process of follicular development is regulated by many factors and granulosa cells (GCs) play an essential role in this process. The proliferation and differentiation of GCs can promote the development and maturation of follicles, and the apoptosis of GCs will lead to follicle atresia [1]. Meanwhile, the steroid hormones produced by GCs are also involved in the growth and development of follicles, including 17 beta-estradiol (E2) and progesterone (P4) [2]. Studies have shown that the level of E2 in follicles is associated with the selection of the dominant follicle [3,4], and higher levels of E2 can increase the luteinizing hormone (LH) and induce ovulation [5]. Levels of P4 that are elevated during pregnancy reduce follicle recruitment [6], whereas low levels of P4 increase follicle diameter [7] and promote early follicular growth and development [8]. Interestingly, the synthesis of E2 and P4 in GCs is regulated by a variety of steroid synthases. The free cholesterol in GCs is transferred to the inner mitochondrial membrane under the action of the steroid synthesis acute regulator (*StAR*) [9], which is then catalyzed by *CYP11A1* to generate pregnenolone. Then, pregnenolone is catalyzed into progesterone by *3β-HSD* [10]. Additionally, *CYP19A1* can convert androstenedione into E2 [11].

The bone morphogenetic protein families (BMPs) are the largest subfamily of the TGF-β growth factor superfamily [12] and they are widely involved in follicular development, steroid generation, ovulation, and luteinization [13], all of which are important regulators of folliculogenesis. In addition, the BMPs’ proteins secreted by the ovary are involved in the formation and function of mammalian germ cells and are widely involved in the regulation of ovarian function [14]. BMP6 is a member of the BMPs and is highly expressed in oocytes and GCs [15]. In human antral follicles, BMP6 is expressed in GCs and oocytes with the highest expression abundance [16]. In rats, BMP6 is mainly expressed in growth follicle GCs but not detected in dominant follicles [17], and the mRNA level of BMP6 in goat primary and secondary follicles is significantly higher than that in the original follicles [18]. These studies show that BMP6 is expressed in the GCs of follicular growth in different species, but that the expression of BMP6 is different among different species.

As a key regulatory factor in the ovary, BMP6 plays an important regulatory role in the process of primary to secondary follicle conversion, dominant follicle selection, and follicle atresia in mammals [19]. The study found that the genetic deletion of BMP6 reduced the ovulation rate and oocyte quality in female mice, reducing litter size [20]. In particular, BMP6 also plays an important role in regulating steroid hormone production. BMP6 treatment reduces the expression of *StAR* but upregulates the expression of *CYP19A1* in human luteinized GCs [21]. In rat GCs, BMP6 decreases the production of P4 by inhibiting the activity of *StAR* and P450 side-chain lyases [22]. However, the role of BMP6 in steroid hormone synthesis in goat ovarian GCs is still unclear. Therefore, in order to explore the function of BMP6 in goat ovarian GCs, the effects of the exogenous addition of BMP6 on the proliferation, apoptosis, cycle, and steroid hormone secretion in goat ovarian GCs were analyzed in this study.

## 2. Materials and Methods

### 2.1. Chemicals and Reagents

DME/F-12 cell culture medium (HyClone, Logan, UT, USA), Fetal bovine serum (HyClone, Logan, UT, USA), Trypsin (HyClone, Logan, UT, USA), FSHR immunohistochemical kit (anti-rabbit) (Xuanya Bio, Shanghai, China), PCNA Rabbit pAb (Bioworld, St. Louis, MN, USA), β-Actin Rabbit pAb (Proteintech, Wuhan, China), Anti-Rabbit IgG (H+L) (Proteintech, Wuhan, China), BeyoClick™ EdU-594 (Beyotime, Shanghai, China), Recombinant Human BMP-6 (R&D Systems, Minneapolis, MN, USA), Annexin V-FITC/PI, Protein Marker (Solarbio, Beijing, China), RNAiso, Prime Script™ RT reagent Kit with gDNA Eraser, and TB Green™ Premix Ex Taq™ Ⅲ (TaKaRa, Kusatsu, Japan).

### 2.2. Cell Cultures and Treatment

From the experimental animal sheep farm of Southwest University, twenty Dazu black goat ewes aged 3–4 months with a strong physique, good mental state, and no reproductive-function-related diseases were selected as the experimental animals in this study. The experiment was conducted in strict accordance with the regulations of the International Committee for Animal Welfare Cooperation (ICCAW) and the Animal Experiment Ethics Committee of Southwest University (Southwest University (2017) No. 7). After the goat was slaughtered, fresh ovaries were collected, stored in sterile saline at 37 °C, and brought back to the laboratory within 1 h. The culture of goat ovarian GCs was the same as that reported by Zhao [23]. The cell culture medium was composed of 90% DME/F-12 and 10% fetal bovine serum (FBS). When the cell density reached 80%, the cell culture medium was discarded and washed twice with phosphate-buffered saline (PBS). Then, the culture medium prepared with the addition of BMP6 was then added and the granulosa cells were cultured in a cell incubator (Thermo, Waltham, MA, USA) at 37 °C and 5% CO_2_.

### 2.3. Cellular Immunochemical Staining

In order to identify whether the cells isolated and cultured in this study were GCs, cell immunostaining was performed using the follicle-stimulating hormone receptor (FSHR), specifically expressed in the GCs’ membranes. The cells to be detected were inoculated into a 96-well plate and cultured in a cell incubator (Thermo, Waltham, MA, USA) at 37 °C and 5% CO_2_ for 24 h. Then, the cells were fixed with 4% paraformaldehyde for 10 min, followed by the addition of 0.5% TritonX-100 for cell permeabilization for 10 min, and then the cells were sealed with goat serum block solution at room temperature for 30 min. Primary antibody (anti-rabbit FSHR) diluted with 1% bovine serum albumin (BSA) solution was added dropwise to the blocked sections and the cells were incubated in a cell incubator (Thermo, Waltham, MA, USA) at 37 °C and 5% CO_2_ for 2 h. The cells were washed three times with sterile PBS (5 min each time), followed by the addition of a secondary antibody diluted with a 1% BSA solution and incubated in a cell culture incubator at 37 °C and 5% CO_2_ for 30 min. After the incubation, the horseradish enzyme-labeled streptavidin solution diluted with a 1% BSA solution was added and the cells were placed in a cell incubator at 37 °C and 5% CO_2_ for 30 min. The freshly prepared DAB working solution was incubated for 5 min at room temperature. Finally, an appropriate amount of hematoxylin staining solution was added to counterstain the cells, which were then incubated at room temperature for 5 min, rinsed with tap water for 5 min, and observed with an inverted fluorescence microscope (Leica, Weztlar, Germany) to observe the staining state of the cells and photograph for preservation.

### 2.4. Ovarian Immunohistochemistry

Immunohistochemistry was used to demonstrate the expression of BMP6 in the goat ovary. After the goats were slaughtered, the obtained ovaries were stored in 4% paraformaldehyde at 4 °C, and after removal, the ovarian tissue was cut to an appropriate size, placed in an embedding box, and fixed with formalin for 24 h. After the fixation, the ovarian tissue was dehydrated, made transparent, and wax-dipped, and then embedded in an embedding machine. Then, it was cut into paraffin sections, which were placed in a constant temperature display machine for display and dried for use. The dried sections were deparaffinized and then placed in a retrieval box containing citric acid antigen retrieval buffer for antigen retrieval and then placed in a 3% hydrogen peroxide solution for incubation at room temperature for 25 min in the dark. After the paraffin sections had dried, 3% BSA was evenly added dropwise to the tissue and blocked at room temperature for 30 min. After blocking, the primary antibody was added dropwise and the sections were incubated overnight at 4 °C in a wet box. Then, the sections were washed three times with PBS (5 min each time), a secondary antibody was added dropwise, and then the sections were incubated at room temperature for 50 min. Then, the sections were washed three times with PBS (5 min each time). Finally, freshly prepared DAB chromogenic solution was added dropwise for color development, and the positive color was brown yellow under microscope observation. After staining, the sections were dehydrated, made transparent, and then sealed with neutral gum after drying.

### 2.5. EdU Cell Proliferation Measurements

The goat ovarian granulosa cells inoculated into 12-well plates were treated with exogenous BMP6 (50 ng/mL) for 48 h. When the cell density reached 50% of the bottom of the culture plate, the culture medium was discarded and an equal volume of EdU-working solution (20 μM) and DME/F-12 were added to each well and incubated for 2 h. After the incubation, the culture medium was discarded, 4% paraformaldehyde fixative was added, and the cells were fixed at room temperature for 15 min. After washing three times with the washing solution, a permeable solution containing 0.3% Triton X-100 was added and the cells were incubated at room temperature for 10–15 min. After washing three times with the washing solution, 200 μL of newly prepared Click reaction solution (Click Reaction Buffer 172 μL, CuSO_4_ 8 μL, Azide 594 0.4 μL, Click Additive Solution 20 μL) was added to each well and the cells were incubated in the dark for 30 min at room temperature. After the Click reaction solution was discarded and washed three times with washing solution, 1 mL of Hoechst 33342 working solution (1×) was added to each well and the cells were incubated in the dark at room temperature for 10 min. Then, the Hoechst 33342 working solution was discarded and the cells were washed three times with the washing solution. Fluorescence detection was performed under an inverted fluorescence microscope (Leica, Weztlar, Germany).

### 2.6. Apoptosis and Cell Cycle Measurements

Apoptosis of granulosa cells was detected by flow cytometry (Beckman Coulter, Brea, CA, USA). After the cells were treated with exogenous BMP6 (50 ng/mL) for 48 h, the cultured cells were digested and suspended to adjust the cell density to 10^7^ cells/mL. An amount of 100 μL of cell suspension was added to a flow cytometry tube, followed by Annexin V (5 μL) and propidium iodide (PI), respectively, and gently rubbed and mixed. The cells were incubated at room temperature in the dark for 15 min, the cell status was detected by flow cytometry, and the percentage of apoptosis was calculated. The cell cycle distribution was determined by flow cytometry. The cells were collected by digestion; 1–5 × 10^6^ cells were collected for each treatment and the cells were resuspended and fixed in 500 μL of 70% pre-chilled ethanol and then kept at 4 °C overnight. Then, 200 μL of rapid cell cycle detection reagent was added and the cells were gently inverted and mixed to prepare a single-cell suspension. Finally, the single-cell suspensions were analyzed for cell cycle by a flow cytometry within 1 h.

### 2.7. Steroid Assay

After the cells were treated with different concentrations of BMP6 (0 ng/mL, 10 ng/mL, 50 ng/mL, or 100 ng/mL) for 24 h or 48 h, the cultured medium of granulosa cells was collected, and the levels of 17 beta-estradiol and progesterone were detected by an ELISA kit (Cayman, Ann Arbor, MI, USA). The ELISA procedure was performed according to ELISA kit instructions, and the optical density (OD) values of each treatment group at 450 nM were measured by a microplate reader (Bio-Rad, Hercules, CA, USA). Then, the concentrations of 17 beta-estradiol (E2) and progesterone (P4) in the medium were calculated by linear regression of the standard curve.

### 2.8. Quantitative Real-Time PCR

After treatments, total RNA was extracted from the GCs by RNAiso reagent (TaKaRa, Kusatsu, Japan), and the concentration and purity of RNA were determined by a spectrophotometer (Thermo, Waltham, MA, USA). Reverse transcription of mRNA into cDNA was performed by a PrimeScript™ RT Reagent kit with gDNA Eraser. The primers and sequences used in this study are shown in Table 1. The quantitative real-time PCR protocol was as follows: TB Green™ Premix Ex Taq™ II 7.5 μL, Forward primer (10 μmol/L) 0.6 μL, Reverse primer (10 μmol/L) 0.6 μL, template (cDNA) 1.2 μL, and RNase Free dH_2_O 5.1 μL. The reaction conditions were performed as follows: 95 °C for 30 s; 95 °C for 5 s; and 60 °C for 30 s for 40 cycles, and the fluorescence signal was detected at 63.5 °C. The melting curve is based on specific primers in the range of 65–95 °C. Adjustments were made to read Ct values every 0.5 °C. The relative expression levels of genes were determined by the 2^ΔΔCt^ method. 

### 2.9. Western Blotting

According to the instructions of the animal whole protein extraction kit (Solarbio, Shanghai, China), the total protein of cells with different treatments was extracted after the cells were treated. The total protein was subpackaged as required and stored in a −80 °C refrigerator (Sanyo, Osaka, Japan). The protein concentration was determined using a BCA protein assay kit (Bioground, Chongqing, China). Approximately 20 µg of protein samples per well were separated on 12% SDS PAGE gels and blotted on a polyvinylidene fluoride (PVDF) membrane (Biosharp, Beijing, China). The proteins were blocked in TBST containing 5% non-fat dry milk with slight shaking for 1 h at room temperature and then washed three times for 10 min with TBST buffer (20 mM Tris-HCl, 150 mM NaCl, 0.05% Tween 20). Then, the membranes were incubated overnight at 4 °C with primary antibodies PCNA Rabbit pAb (1:1000; Bioworld, St. Louis, MN, USA) and β-Actin Rabbit pAb (1:5000; Proteintech, Wuhan, China). After washing with TBST buffer, the membranes were incubated with secondary antibody anti-rabbit (1:2000; Proteintech, Wuhan, China) at room temperature for 2 h. After washing at least three times, the protein bands were visualized with ECL (Bioground, Chongqing, China).

### 2.10. Statistical Analysis

In this experiment, Prism software (GraphPad, San Diego, CA, USA) was used for statistical analysis. An independent *t*-test was used for comparative analysis of the differences between the two groups of data, and multiple comparisons were used for the differences between more than two groups. All experiments were repeated at least three times, and the results were expressed as “mean ± standard error (SEM)”. *p* < 0.05 indicated a significant difference, whereas *p* > 0.05 indicated a non-significant difference.

## 3. Results

### 3.1. Identification of Goat Ovarian Granulosa Cells and Expression of BMP6 in Ovaries

The follicle-stimulating hormone receptor (FSHR) is specifically expressed on the membrane of ovarian GCs. The cells isolated and cultured can be identified as GCs by cell immunostaining (Figure 1A–C). The GC membrane was colored yellow and brown and the nucleus was colored blue (Figure 1C). These results showed that the GCs were successfully isolated and cultured in this experiment; in addition, they could be used in subsequent experiments. After immunostaining, it was observed that BMP6 was widely distributed in goat ovaries and that BMP6 was abundantly expressed in the GCs (Figure 1D).

### 3.2. Effects of BMP6 on Proliferation in GCs

The EdU staining showed that the percentage of EdU-positive cells in the BMP6-treated group was not significantly different from that in the control group after the granulosa cells were treated with exogenous BMP6 (50 ng/mL) for 48 h (*p* > 0.05; Figure 2A,B). Then, the mRNA and protein expression levels of the proliferation-related gene *PCNA* were quantitatively analyzed by qRT-PCR assays and Western blotting. The results showed that compared with the control group, the BMP6 treatment group had no significant effect on the expression of *PCNA* (*p* > 0.05; Figure 2C,D). The results showed that the exogenous BMP6 treatment did not affect the proliferation of GCs in vitro.

### 3.3. Effects of BMP6 on Cell Cycle in GCs

Compared with the control group, the results showed that the exogenous BMP6 (50 ng/mL) treatment for 48 h had no significant effect on the periodic distribution of GCs (*p* > 0.05; Figure 3A,B). The mRNA expression levels of the cell cycle-related genes *CDK4*, *CCNE1*, and *CCND1* were quantitatively analyzed by qRT-PCR assays. As shown in Figure 3C, BMP6 treatment increased the mRNA expression of *CDK4* (*p* < 0.05) and *CCND1* (*p* < 0.01), whereas the mRNA expression of *CCNE1* was significantly decreased by BMP6 treatment (*p* < 0.01; Figure 3C). These results showed that the exogenous BMP6 treatment did not change the cell cycle progression of GCs in vitro but could affect the expression of cell cycle-related genes.

### 3.4. Effects of BMP6 on Apoptosis in GCs

The results showed that the exogenous BMP6 treatment had no significant effect on the apoptosis rate of GCs compared with the control group (*p* > 0.05; Figure 4A,B). Then, the mRNA expression levels of the apoptosis-related genes were quantitatively analyzed by qRT-PCR assays. Compared with the control group, the exogenous BMP6 treatment had no significant effect on the mRNA expression levels of apoptosis-related genes *BAX*, *BCL2,* and *Caspase3* (*p* > 0.05; Figure 4C). The results showed that the exogenous BMP6 treatment did not affect the apoptosis of GCs in vitro.

### 3.5. Effects of BMP6 on Steroid Hormones’ Secretion in GCs

In order to investigate the effect of BMP6 on steroid hormone synthesis in goat ovarian GCs in vitro, the concentrations of E2 and P4 in a serum-free medium were detected by an ELISA kit. The results showed that whether the exogenous BMP6 was treated for 24 h or 48 h, the levels of E2 and P4 were upregulated in a BMP6 concentration-dependent manner (*p* < 0.01; Figure 5A–D). The mRNA expression levels of the steroid hormone synthesis-related genes were quantitatively analyzed by qRT-PCR assays. As shown in Figure 5E, BMP6 treatment significantly increased the mRNA expression of *CYP11A1* and *CYP19A1* and significantly decreased the mRNA expression of *3β-HSD* and *StAR* (*p* < 0.01). In addition, BMP6 treatment significantly reduced the mRNA expression of *ESR1* (*p* < 0.01; Figure 5F) but did not affect the mRNA expression levels of *LHR* and *FSHR* (*p* > 0.05; Figure 5F). These results showed that BMP6 treatment significantly increased the secretion of E2 and P4 and the mRNA expression of E2 and P4 rate-limiting enzymes in GCs.

## 4. Discussion

BMP family members are considered key regulatory factors, which are widely involved in cell proliferation, differentiation, apoptosis, and embryogenesis [24]. An in vitro study found that BMP6 could increase the maturation of antral follicles in mice [25]. Meanwhile, the exogenous addition of BMP6 protein to the GCs of chicken ovary cultured in vitro significantly promoted the proliferation of the GCs [26]. In this experiment, BMP6 treatment had no significant effect on proliferation in goat ovary GCs in vitro. Additionally, a study has found that BMP6 does not affect the proliferation of GCs in a rat model [22]. These results demonstrate that there may be differences in the effects of BMP6 treatment on the proliferation of GCs in mammals and poultry.

In lens cells, BMP signaling will arrest the cell cycle in the S phase, thereby regulating the process of cell proliferation [27]. In breast cancer cells, BMP6 significantly inhibits cell proliferation by reducing the number of cells in the S phase of the cell cycle, thereby blocking tumorigenesis [28]. However, the present study found that BMP6 treatment had no significant effect on the cell cycle progression of goat ovarian GCs. The cell cycle is a highly sophisticated and complex process. For example, cyclin-dependent kinase 4 (*CDK4*) can regulate the G1-S cell cycle transition and trigger gene expression that promotes the entry of the S phase [29]. However, there is also a study showing that *CDK4* is dispensable for G1/S progression in normal development [30]. *CCND1* is a gene that encodes the core component of the cell cycle mechanism, which helps trigger cell transition from the G1 to the S phase [31]. *CCNE1* is also an active regulator of cell cycle regulation, which promotes the G1 to S phase transition by binding to and activating *CDK2* [32]. All three of the above genes can promote the transition of cells from the G1 phase to the S phase. In this experiment, BMP6 treatment upregulated the mRNA expression of *CDK4* and *CCND1* but downregulated the expression of *CCNE1*. Although BMP6 treatment regulated the expression of these genes closely related to the cell cycle, BMP6 had no significant effect on the progression of the cell cycle and the mechanism remains to be further elucidated. 

During ovarian development and maturation in mammals, a complex spontaneous phenomenon called follicular atresia occurs, which affects the various stages of follicular growth and development. Follicular atresia is usually caused first by the apoptosis of GCs [33]. The paracrine factors of oocytes can effectively inhibit the apoptosis of cumulus cells [34], whereas in this experiment, the exogenous BMP6 treatment had no significant effect on the apoptosis rate of ovarian GCs. It has been reported that treating cumulus cells with oocyte exocrine proteins, such as BMP15 and GDF9, or co-treating with BMP15/GDF9, could promote the proliferation of cumulus cells and prevent cell apoptosis within a certain concentration range. However, exogenous concentration changes in BMP15 and GDF9 inhibited the proliferation of cumulus cells and triggered apoptosis [35]. These results indicate that apoptosis is correlated with exogenous BMP6 concentration.

The maturation and ovulation of the oocytes are dependent on the steroid hormones secreted by GCs in the follicle, and estradiol and progesterone play an important regulatory role in this process [36]. E2 can reduce the apoptosis rate and promote the proliferation in GCs, but high concentrations of E2 can inhibit the proliferation [37,38]. P4 can slow down follicular development, inhibit the mitosis and apoptosis of granulosa cells, maintain the granulosa cell state, balance follicle numbers, and improve ovarian reserve [39,40]. In this study, BMP6 treatment significantly promoted the secretion of E2 and P4 in GCs after 24 h or 48 h, suggesting that BMP6 is involved in regulating follicle development by regulating steroid secretion. Mammalian steroid production begins with the *StAR*-mediated rate-restricted transport of cholesterol to the mitochondria, where cholesterol is catalyzed by *CYP11A1* [41] and converted into progesterone, the first precursor in the steroidogenic cascade [42]. After progesterone is generated, it is converted into other intermediates by *3β-HSD* [43], and *CYP19A1*. As one of the rate-limiting enzymes of E2 biosynthesis, *CYP19A1* converts intermediates into E2 in subsequent steps [44]. Additionally, the BMP family is involved in the regulation of steroid synthesis, and BMP4 inhibits P4 production by inhibiting the expression of *StAR* in bovine GCs [45]. Meanwhile, BMP15 also reduces the expression of *StAR* in human GCs, thereby inhibiting P4 production [46]. In human ovarian GCs, BMP15 also promotes the expression of *CYP19A1* and induces E2 production [47]. In this experiment, we found that BMP6 treatment significantly increased the mRNA expression of *CYP11A1* and *CYP19A1* and significantly decreased the mRNA expression of *3β-HSD* and *StAR*, indicating that BMP6 promotes the secretion of E2 and P4 by increasing the expression of rate-limiting enzymes. In addition, E2 can stimulate the growth and development of follicles to maturity by binding to *ESR1* on the cell membrane surface [48]. In this study, BMP6 could downregulate the mRNA expression of *ESR1*, suggesting that BMP6 may be involved in blocking the development of follicles by inhibiting the sensitivity of granulosa cells to estrogen, thereby regulating ovarian reserve function.

## 5. Conclusions

In summary, the results of this study showed that BMP6 did not affect the proliferation, cell cycle progression, and apoptosis of goat ovarian GCs, but upregulated the mRNA expression of cell cycle-related genes *CDK4* and *CCND1* and downregulated the mRNA expression of *CCNE1*. Importantly, BMP6 promoted the secretion of E2 and P4 via upregulating the expression of rate-limiting enzymes *CYP11A1* and *CYP19A1*.

## Figures and Tables

**Figure 1 animals-12-02132-f001:**
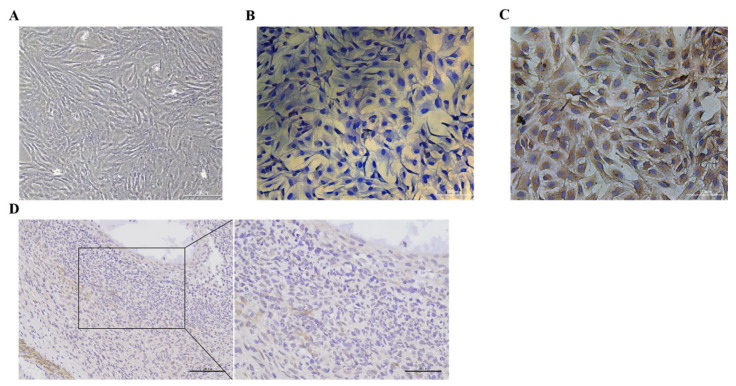
Isolation, culture, and identification of goat ovary GCs and expression of BMP6 in the goat ovary. (**A**) Growth morphology of goat ovary GCs. (**B**) Cell immunostaining of the negative control group using anti-rabbit IgG as the primary antibody. (**C**) Cell immunostaining using the FSHR antibody. The granulosa cell membrane was stained yellow brown and the nucleus was stained blue by DAPI. Bar: 100 μm. (**D**) The expression of BMP6 in the goat ovary was detected by immunohistochemistry, and the immunospecific staining was brown as in the BMP6-positive cells. Bars: 100 μm and 50 μm.

**Figure 2 animals-12-02132-f002:**
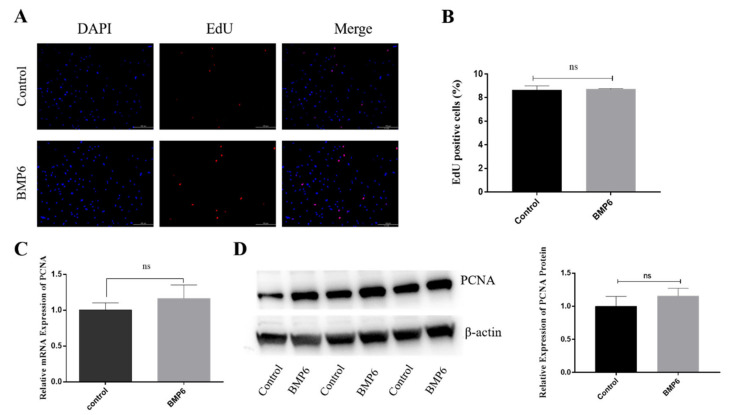
Exogenous BMP6 treatment did not affect the proliferation of GCs. (**A**,**B**) Edu immunofluorescence identification and proportion of positive cell analysis. (**C**) The mRNA level of the proliferation-related gene *PCNA* was detected by qRT-PCR. (**D**) The protein levels of the PCNA were detected by Western blotting. Bar: 100 μm. Data are shown as the mean ± SEM of three independent repeated experiments. β-actin was used as the reference gene for qRT-PCR and Western blotting. NS indicated no significant difference (*p* > 0.05).

**Figure 3 animals-12-02132-f003:**
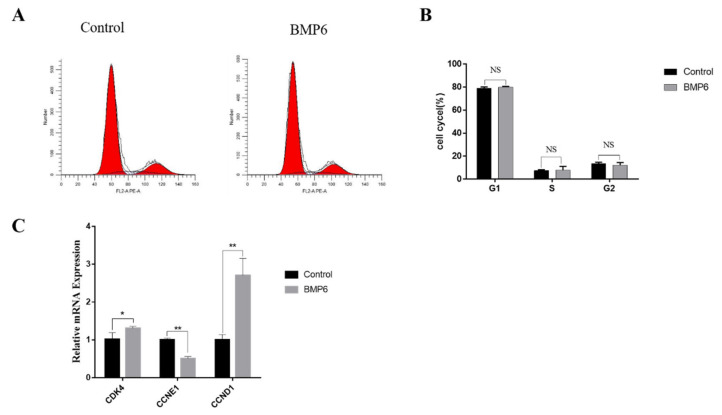
Exogenous BMP6 treatment did not affect the cell cycle progression of GCs but affected the expression of cell cycle-related genes. (**A**,**B**) Flow cytometry was used to detect and analyze the distribution ratio of the GC cell cycle. (**C**) The mRNA expression levels of the cell cycle-related genes (*CDK4*, *CCNE1*, and *CCND1*) were detected by qRT-PCR. Data are shown as the mean ± SEM of three independent repeated experiments. β-actin was used as the reference gene. NS indicated no significant difference (*p* > 0.05). * *p* < 0.05 indicates a significant difference, ** *p* < 0.01 indicates an extremely significant difference.

**Figure 4 animals-12-02132-f004:**
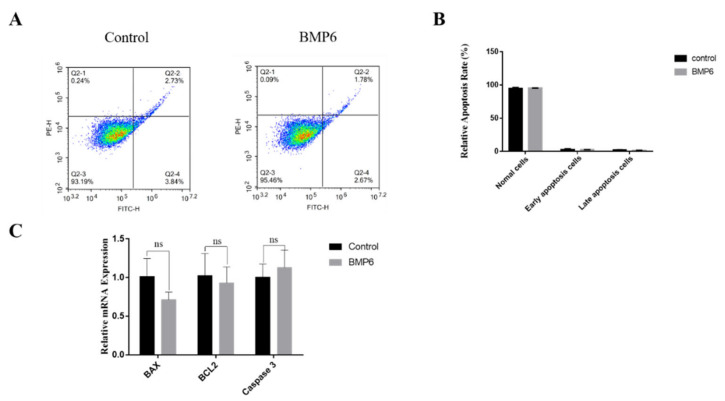
Exogenous BMP6 treatment did not affect the apoptosis of GCs. (**A**) The apoptosis rate of granulosa cells was detected by flow cytometry. The Q2-1 area represents the cell debris after apoptosis, the Q2-2 area represents the cells in the late apoptotic stage, the Q2-3 area represents the normal growing cells, and the Q2-4 area represents the cells in the early apoptotic stage. (**B**) Analysis of the proportion of cells in the apoptosis stage. (**C**) qRT-PCR was used to determine the levels of mRNA expression of cell cycle-related genes (*BAX*, *BCL2,* and *Caspase3*). Data are shown as the mean ± SEM of three independent repeated experiments. β-actin was used as the reference gene. NS indicated no significant difference (*p* > 0.05).

**Figure 5 animals-12-02132-f005:**
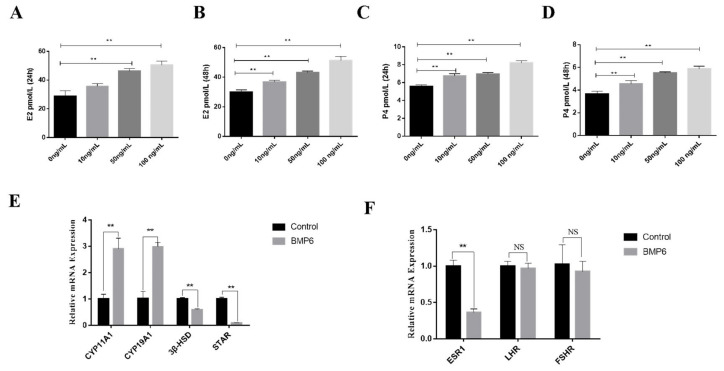
Exogenous BMP6 treatment promotes the secretion of steroid hormones in GCs. (**A**,**B**) The level of 17 beta-estradiol (E2) was detected by an ELISA kit in a GC culture medium after 24 h or 48 h of treatment. (**C**,**D**) The level of progesterone (P4) was detected by an ELISA kit in a GC culture medium after 24 h or 48 h of treatment. (**E**) The mRNA levels of steroid hormone synthesis-related genes (*CYP11A1*, *CYP19A1*, *3β-HSD*, and *StAR*) were analyzed by qRT-PCR assays. (**F**) The mRNA levels of steroid hormone binding receptor-related genes (*ESR1*, *LHR*, and *FSHR*) were analyzed by qRT-PCR assays. Data are shown as the mean ± SEM of three independent repeated experiments. β-actin was used as the reference gene. NS indicated no significant difference (*p* > 0.05). ** *p* < 0.01.

**Table 1 animals-12-02132-t001:** Sequences for primers used in quantitative real-time RT-PCR.

Gene Name	Forward Primer (5′→3′)	Reverse Primer (5′→3′)
BAX	CCAAGAAGCTGAGCGAGTGTCTG	GTGTCCACGGCTGCGATCATC
BCL2	TGTGGATGACCGAGTACCTGAACC	GCCAGACTGAGCAGTGCCTTC
Caspase3	ATACCAGTTGAGGCAGAC	TTAACCCGAGTAAGAATGT
PCNA	GTAGCCGTGTCATTGCGACTCC	GCTCTGTAGGTTCACGCCACTTG
StAR	GCGACCAAGAGCTTGCCTATATCC	TTGGCCTGCCGACTCTCCTTC
CYP19A1	AGGTCATCCTGGTCACCCTTCTG	CGGTCGCTGGTCTCGTCTGG
3β-HSD	CTCAGACGACACACCACACCAAAG	CAGCAGGAAGGCAAGCCAGTAC
CYP11A1	GCTGCGGAAGGAGGTTCTGAATG	GCACCAGTGTCTTGGCAGGAATC
CDK4	GCTGCTGCTGGAGATGCTGAC	CTCTGCGTCACCTTCTGCCTTG
CCND1	TTCCTCTCCTATCACCGCCTGAC	TCCTCTCTTCCTCCTCCTCCTC
CCNE1	AAGTGCTCCTGCCTCAGTATCCTC	ATACAAGGCGGAAGCAGCAAGTAC
ESR1	CTGCTGCTGGAGATGCTGGATG	GCTGGCTCTGATTCACGTCTTCC
LHR	ATTCCGCCATCTTTGCTGAGAGTG	AGCATCTGGTTCAGGAGCACATTG
FSHR	TTTGTGGTCATCTGTGGCTGCTAC	CGCTTGGCTATCTTGGTGTCACTAG
β-actin	TGATATTGCTGCGCTCGTGGT	GTCAGGATGCCTCTCTTGCTC

## Data Availability

The data used in the analysis can be obtained from the authors on request.

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
