# Peer review of "BMP6 Promotes the Secretion of 17 Beta-Estradiol and Progesterone in Goat Ovarian Granulosa Cells"

_animals, 2022, doi:10.3390/ani12162132_

Round 1

Reviewer 1 Report

The manuscript is interesting and has potential in reproductive biotechnics in ruminants. Adequate methods of laboratory analysis were used in the study. Nevertheless, not all details of the study were described, and some important analyses were omitted.

Materials and Methods

How many animals were used for the experiments (N numer) and how many repeats of each experiment was conducted?

In case of ewes, the stage of the reproductive phase is important because ovary can be sensitive or not for steroid action, was was the stage?

Why the authors omit progesterone receptors in this study, whereas estradiol receptors expression was included?

What was the basic for selection of BMP6 dose for eperiments?

Disscusion

The increase of CYP19A1 and CYP11A1 might will connected with sensiticvity of granulosa cells for steroids. The resuls showed decrease ESR1 mRNA expression after BMP6 treatment. How the authors explain such result? This point is not disscussed at all.

Author Response

On behalf of my co-authors, we thank you very much for giving us an opportunity to revise our manuscript. We have studied reviewer’s comments carefully and have made revision which marked in red in the paper. 

Reviewer 2 Report

The paper "BMP6 Promotes the Secretion of 17 beta-Estradiol and Progesterone in Goat Ovarian Granulosa Cells" evaluates the effect of human-BMP6 at different concentration on in vitro cultured goat GCs on proliferation,apoptosis, mRNA expression and steroid hormones synthesis.
The paper is well structured and only few minor english spell check will be necessary.
The are some concerns in M&M and results description, as commented below:

line 94: how many animals were involved? can the authors provide an ethics statement/approval number (I found no specific section at the end of the manuscript).   

line 104: did cells incubated for 24 and 48 hours constitute two different treatment groups? in this case, you have 2 (time) x 4 (BMP6 concentration) = 8 treatment groups and results/discussion shoul be amended accordingly.

paragraphs 2.3 and 2.4: I suggest to move the description of the aim of this sections (which I can read in Results, for example lines 210-212) at the beginning of the 2.3 and 2.4 paragraphs, so the reader will immediately know why the described methods are aimed to. Then, in the results section, author can briefly repeat the concept.

line 207: please, replace "insignificant" with "non-significant".

paragraphs 3.2, 3.3 and 3.4: the author state that at least (see previous comment on incubation time), four treatment groups were created (0 - 10 - 50 -  ng/ml BMP6). Some results, however, consider only the comparison between control and X-concentration of BMP6 treated cells. Please clarify why some results consider only a unique treatment or amend results accordingly.

line 297: "which means and extremely significant difference" please delete this, as it is unnecessary.

line 328-330: it is ok, but results related to apoptosis, for example, are shown as relative to a unique treatment group?

line 330-334: those two sentences are unclear. maybe is there some typo which needs to be fixed in order to improve the meaning of the sentences?

line 336: I suggest to change "depended" with "dependent".

Author Response

(The authors gave the same response as above.)

Round 2

Reviewer 1 Report

In my opinion, the manuscript has been sufficiently improved to warrant publication in Animals.

Reviewer 2 Report

The Authors revised the manuscript according to previous observations and added missing informations as requested.